# Patient and public involvement in numerical aspects of trials: a mixed methods theory-informed survey of trialists' current practices, barriers and facilitators

Beatriz Goulao [ID] , Camille Poisson, Katie Gillies [ID]

► Prepublication history and additional materials for this paper are available online. To view these files, please visit the journal online (http://dx.doi.org/10.1136/bmjopen-2020-046977).

Health Services Research Unit, University of Aberdeen, Aberdeen, UK

**Correspondence to**
Dr Beatriz Goulao;
beatriz.goulao@abdn.ac.uk

## ABSTRACT

**Objective** We aimed to find out if trialists involve patients and the public in numerical aspects of trials, how and what are the barriers and facilitators to doing it.

**Design** We developed a survey based on the Theoretical Domains Framework. We used a mixed methods approach to analyse the data and to identify important domains.

**Setting** Online survey targeting UK-based trial units.

**Participants** Stakeholders working in UK-based clinical trials, 18 years old or over, understand English and agree to take part in the study.

**Outcome measures** Trialists' behaviour of involving patients and the public in numerical aspects of trials and its determinants.

**Results** We included 187 respondents. Majority were female (70%), trial managers (67%) and involved public and patient partners in numerical aspects of trials (60%). We found lack of knowledge, trialists' perception of public and patient partners' skills, capabilities and motivations, scarce resources, lack of reinforcement, and lack of guidance were barriers to involving public and patient partners in numerical aspects of trials. Positive beliefs about consequences were an incentive to doing it.

**Conclusions** More training, guidance and funding can help trialists involve patient and public partners in numerical aspects, although they were uncertain about public and patient partners' motivation to be involved. Future research should focus on identifying public and patient partners' motivations and develop strategies to improve the communication of numerical aspects.

## BACKGROUND

Patient and public involvement (PPI) has been defined as 'research being carried out 'with' or 'by' members of the public (including patients and carers) rather than 'to', 'about' or 'for' them' (http://www.invo.org.uk/). PPI activity has become increasingly common in trials in the UK for several reasons: its supporters believe it makes research more relevant and it is, morally, the right thing to do. It is also a requirement

## Strengths and limitations of this study

► This is the first investigation of trialists' practice regarding involving patients and the public in numerical aspects of trials and aimed at UK-based trialists.
► We achieved a wide representation of trialists working in clinical trials in the UK.
► We described current practice as well as determinants of behaviour using a theory-informed approach based on the Theoretical Domains Framework.
► Respondents' interpretation of involving patients and the public in numerical aspects varied, so the results apply to a spectrum of behaviours.
► Respondents are likely to have stronger views of patient and public involvement (PPI) and PPI in numerical aspects of trial since they opted to answer our survey.

from major funders, including the National Institute for Health Research.[1]

In a trial context, patient and public partners have been primarily involved in agenda setting, steering committees, ethical review, protocol development and piloting,[2] but PPI remains seen as tokenistic and a ticking box exercise in many settings.[3,4] Trials are primarily quantitative research and often key decisions are based on numerical aspects and their discussion. However, the extent to which patients and the public get involved in those discussions is unknown, with some evidence suggesting communication of statistical aspects of trials needs to be clearer in order to achieve meaningful involvement.[5] At the same time, involving patients and the public in the statistical and health economics aspects of trials can result in enriching data interpretation and better research, with more robust and implementable evidence.[6,7]

PPI can be considered a professional behaviour since it involves a series of actions

to accomplish a goal (such as inviting a patient partner to contribute to a discussion). To understand a specific behaviour and its determinants, it can be helpful to use behavioural theory to provide a systematic and replicable framework. The Theoretical Domains Framework (TDF) was developed to focus on healthcare professional behaviour, specifically around evidence-based medicine and the adoption of new practices, and is made of up of 15 behavioural domains.[8] Previous work has used the TDF to explore researchers' beliefs related to PPI in health research,[3] but to our knowledge this is the first application of the framework to explore PPI in trials, and in particular PPI in numerical aspects of trials. We aimed to use a theory-informed approach to find out whether trialists involve patient and public partners in numerical aspects of trials and what are the barriers and facilitators to doing it.

## METHODS

### Survey design

The survey was divided into the following sections:
► Demographic questions.
► Definition of numerical aspects of trials.
► General questions about PPI in numerical aspects of trials.
► TDF questions.

Demographic questions focused on characterising our sample (age, gender, trial work experience and role). We agreed on the definition of 'numerical aspects of trials' ('any aspects of a trial where people measure (or plan to measure), manage, analyse or share information that is presented as one or more numbers') through an iterative process involving 13 members of the Health Services Research Unit at the University of Aberdeen (including a variety of different roles in the unit, such as a professor of health services research, a health economist, a statistician, trial manager, PPI/engagement coordinator, research fellow, assistant researcher and quality assurance manager). The definition was included in the survey. Respondents were asked about their current practice regarding involving public and patient partners in numerical aspects of trials.

The TDF was developed as a theoretical framework and includes behavioural theories and constructs and proposes that determinants of professionals' behaviour can be clustered in 'domains'. The TDF, after an update from Presseau et al,[9] consists of 15 domains. The TDF section of our survey was developed based on previous surveys[10 11] and was reviewed by the project team and by an expert in TDF research (ED).

The survey was piloted by eight staff or student members from the University of Aberdeen. The survey was administered and delivered through SurveyMonkey. Informed consent was requested and embedded in the survey. The final version contained 43 questions consisting of both closed and open questions (see online supplemental appendix 1 – questionnaire). The participants were able to withdraw from the survey at any time.

### Target population

The target population was any stakeholder working in clinical trials (from here onwards called a 'trialist'), for example trial managers, principal or chief investigators, statisticians, clinicians, patient or public partners, PPI coordinators, and research fellows. To take part in our study, participants needed to be 18 years old or over, work in a UK clinical trial's unit, understand English and agree to take part in the study.

### Participant recruitment

The questionnaire was disseminated using an online link via email and social media. We used a snowball approach to recruit survey respondents by disseminating the survey to colleagues, known networks and through social media (LinkedIn, Twitter, Facebook and Instagram). We used common hashtags (#) and pictures to encourage responses. We contacted stakeholders via professional mailing lists to ask to disseminate our survey, such as a PPI coordinator's mailing lists, clinical trial unit websites and the UK Clinical Research Collaboration. The survey link was active from 11 June 2019 to 1 July 2019.

### Survey analysis

The survey analysis was divided into two parts: analysis of closed-ended questions (as quantitative data) and analysis of open questions (as qualitative data). We exported the data into Stata V.15 and generated statistical summaries of the closed-ended questions. The Likert scale questions used a scale from 1 (weaker agreement/confidence/etc) to 7 (stronger agreement/confidence/etc). They are presented using median, IQR and counts. Binary questions (with only two possible answers; eg, yes/no) are presented as n (%) out of the available answers. We split the results into respondents who self-reported to perform the behaviour assessed (involving public and patient partners in numerical aspects of trials) and those who did not or were not sure. When questions included a category named 'other', the answers given were grouped together, coded and counted. Analysis of open-ended questions was done using a deductive thematic analysis approach informed by the TDF by one author (BG). If data did not fit into the prespecified TDF domains, an inductive approach was taken. We compared the quantitative and qualitative data to identify the most salient domains of the TDF that explained the behaviour. This process was done by one author (BG) and reviewed using a subset of qualitative data by a second author (KG). We did not undertake any statistical testing since our aim was to describe the data and not to test hypotheses. We used Stata V.15 for the purposes of summarising quantitative data and Microsoft Word to analyse qualitative data.

## Identifying the most 'relevant' domains to involve patients and the public in numerical aspects of trials

In order to establish the barriers and facilitators of the target behaviour (PPI in numerical aspects of trials), criteria were developed to determine which domains of the TDF were 'relevant' for this behaviour based on previous TDF research.[12] Relevant domains were those that were considered to show variation in reported behaviour/beliefs between trialists who involve patients and the public in numerical aspects and those who do not.

### PPI in the study

Our study is part of a larger project, the Patient and public Involvement in Numerical aspects of Trials (PoINT) project (https://www.abdn.ac.uk/hsru/what-we-do/research/projects/point-827.php), where PPI has been sought at all stages including in its design and via focus groups. The current survey was piloted by three public members.

## RESULTS

A total of 317 people accessed the survey. From these 317, we excluded from the final analysis 130 people for the following reasons: 31 did not provide any information and 99 did not respond to any TDF-focused questions. Therefore, the 187 participants who replied to at least one TDF question were included in the final analysis (figure 1). Online supplemental table 1 provides the demographic information based on respondent status. Respondents and non-respondents had similar features with a few exceptions: there was a higher percentage of women in the group of non-respondents compared with respondents (n=82, 83% vs n=130, 70%), and there was a higher percentage of respondents involved in all stages of trials (n=75,

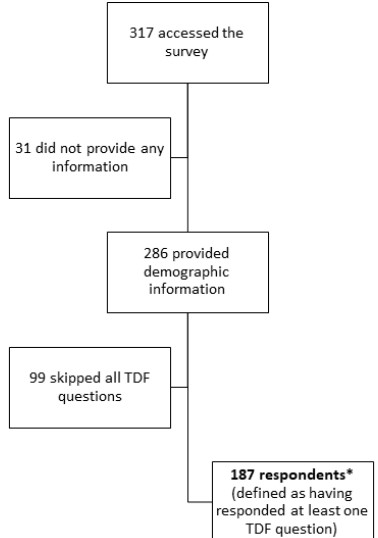

**Figure 1** Flow of participants. *Demographic information by respondent status is presented in online supplemental table 1. TDF, Theoretical Domains Framework.

40%) compared with non-respondents (n=25, 25%). However, the most common trial stages that respondents and non-respondents were involved in were the same: trial design (around 50%) and undertaking the trial (around 50%). There was a higher proportion of statisticians in the respondents group (n=29, 16%) compared with non-respondents (n=8, 8%). A larger percentage of respondents reported to involve patient and public partners in numerical aspects of trials daily compared with non-respondents (n=48, 26% vs n=11, 11%).

### Characteristics of respondents

Table 1 shows the characteristics of the respondents. Of the respondents, 70% were women (n=130). Respondents were based across the UK, with the highest percentage coming from London (n=33, 18%). Around 40% were involved in all stages of the trial (n=75) and about half were involved in design, conduct and dissemination. One-third were involved in the analysis of the trial. Most respondents were trial managers (n=125, 67%); eight respondents (4%) were patient partners in trials. About 37% (n=69) had been working in trials for more than 10 years and 31% for less than 5 (n=57).

### Respondents' current practice of involving public and patient partners in numerical aspects of trials

Around 62% of respondents said they involve public and patient partners in numerical aspects of trials (n=116); 26% said they did not (n=48) and 12% said they were not sure (n=23). From the trialists involving public and patient partners in numerical aspects of trials, 24% have been working in the field for less than 5 years (n=28) vs 41% who did not involve public and patient partners in numerical aspects of trials (n=29). Some respondents were uncertain about what involving public and patient partners in numerical aspects meant, with definitions ranging from any discussion about numbers to involvement in the statistical analysis.

The most common stages of trials to involve public and patient partners in numerical aspects were trial design (n=86, 74%) and dissemination (n=84, 72%). About half involved public and patient partners in the analysis and interpretation. Respondents exemplified how they involved public and patient partners in numerical aspects: in management and steering committees where numerical data are discussed; sample size calculations, and specifically discussing effect sizes, clinically important differences and non-inferiority margins; dissemination of results and aiding to increase clarity of communication with trial participants; grant applications and trial design stage; when and how to collect data; and defining outcomes. Comments made about involving public and patient partners in the statistical analysis came from those

**Table 1** Descriptive statistics by trialists' current practice (involving PPP in numerical aspects of trials or not/not sure) and in total

| | Trialists who involve PPP in numerical aspects (n=116) | Trialists who do not involve PPP in numerical aspects or do not know (n=71) | Total |
|---|---|---|---|
| Age (years) | | | |
| 18–24 | 1 (0.9) | 1 (1.4) | 2 (1.1) |
| 25–34 | 32 (27.6) | 30 (42.3) | 62 (33.2) |
| 35–44 | 38 (32.8) | 12 (16.9) | 50 (26.7) |
| 45–54 | 25 (21.6) | 16 (22.5) | 41 (21.9) |
| 55–64 | 14 (12.1) | 8 (11.3) | 22 (11.8) |
| 65+ | 5 (4.3) | 2 (2.8) | 7 (3.7) |
| Rather not say | 1 (0.9) | 2 (2.8) | 3 (1.6) |
| Gender | | | |
| Female | 80 (69.0) | 50 (70.4) | 130 (69.5) |
| Male | 33 (28.4) | 17 (23.9) | 50 (26.7) |
| Other | | 2 (2.8) | 2 (1.1) |
| Prefer not to say | 3 (2.6) | 2 (2.8) | 5 (2.7) |
| Work location | | | |
| England | 9 (7.8) | 8 (11.3) | 17 (9.1) |
| North East England | 4 (3.4) | 3 (4.2) | 7 (3.7) |
| North West England | 7 (6.0) | 7 (9.9) | 14 (7.5) |
| Yorkshire and the Humber | 8 (6.9) | 5 (7.0) | 13 (7.0) |
| West Midlands | 10 (8.6) | 2 (2.8) | 12 (6.4) |
| East Midlands | 3 (2.6) | 4 (5.6) | 7 (3.7) |
| South West England | 9 (7.8) | 2 (2.8) | 11 (5.9) |
| South East England | 13 (11.2) | 6 (8.5) | 19 (10.2) |
| London | 19 (16.4) | 14 (19.7) | 33 (17.6) |
| East England | 3 (2.6) | 1 (1.4) | 4 (2.1) |
| Northern Ireland | 3 (2.6) | 3 (4.2) | 6 (3.2) |
| Scotland | 18 (15.5) | 12 (16.9) | 30 (16.0) |
| Wales | 10 (8.6) | 4 (5.6) | 14 (7.5) |
| Which aspects of the trial process are you directly involved in? | | | |
| Trial design | 51 (44.0) | 37 (52.1) | 88 (47.1) |
| Undertaking the trial | 48 (41.4) | 37 (52.1) | 85 (45.5) |
| Dissemination | 55 (47.4) | 34 (47.9) | 89 (47.6) |
| Analysis | 36 (31.0) | 23 (32.4) | 59 (31.6) |
| Overseeing committee | 1 (0.9) | 4 (5.6) | 5 (2.7) |
| All stages | 56 (48.3) | 19 (26.8) | 75 (40.1) |
| Main role in trial | | | |
| Trial manager | 81 (69.8) | 44 (62.0) | 125 (66.8) |
| Chief investigator | 2 (1.7) | 1 (1.4) | 3 (1.6) |
| Statistician | 16 (13.8) | 13 (18.3) | 29 (15.5) |
| Health economist | 2 (1.7) | 1 (1.4) | 3 (1.6) |
| Qualitative researcher | 6 (5.2) | 2 (2.8) | 8 (4.3) |
| Patient partner | 4 (3.4) | 4 (5.6) | 8 (4.3) |
| Coinvestigator | 9 (7.8) | 7 (9.9) | 16 (8.6) |

Continued

**Table 1** Continued

| | Trialists who involve PPP in numerical aspects (n=116) | Trialists who do not involve PPP in numerical aspects or do not know (n=71) | Total |
|---|---|---|---|
| Researcher | 6 (5.2) | 1 (1.4) | 7 (3.7) |
| Systematic reviewer | | 1 (1.4) | 1 (0.5) |
| PPI lead | 1 (0.9) | 2 (2.8) | 3 (1.6) |
| Not reported | 1 (0.9) | | 1 (0.5) |
| How long have you been working in trials? | | | |
| Less than 5 years | 28 (24.1) | 29 (40.8) | 57 (30.5) |
| Between 5 and 10 years | 44 (37.9) | 17 (23.9) | 61 (32.6) |
| More than 10 years | 44 (37.9) | 25 (35.2) | 69 (36.9) |
| When do you involve PPP in numerical aspects of trials? | | | |
| Never | 2 (1.7) | | 2 (1.1) |
| Rarely | 10 (8.6) | | 10 (5.3) |
| Regularly | 72 (62.1) | | 72 (38.5) |
| Every day | 32 (27.6) | | 32 (17.1) |
| What stage of a trial do you involve PPP in the numerical aspects? | | | |
| Trial design | 86 (74.1) | | 86 (46.0) |
| Undertaking the trial | 58 (50.0) | | 58 (31.0) |
| Dissemination | 84 (72.4) | | 84 (44.9) |
| Undertaking the trial | 58 (50.0) | | 58 (31.0) |
| Overseeing committee | 8 (6.9) | | 8 (4.3) |
| No information available | 2 (0.9) | | 4 (1.6) |

Values are n (%).
PPI, patient and public involvement; PPP, patient and public partner.

who did not see an interest from public and patient partners in getting involved. For example:

> I've also yet to have any PPI members say they would like to be more involved in this side of things; except for post-analysis when considering the qualitative and quantitative aspects together. (Trial manager and statistician, involves)

### Domains influencing trialists in involving patients and the public in numerical aspects of trials

TDF domains selected as the most relevant for the target behaviour are presented in table 2; all TDF domain results are presented in online supplemental table 2. Table 3 summarises the themes identified from the responses to open questions. By comparing the quantitative and qualitative data, we identified knowledge, skills, beliefs about capabilities, beliefs about consequences, reinforcement, environmental context and resources, social influences and behavioural regulation as being salient TDF domains. We identified an additional theme within the data on 'trial communication culture' that did not fit within the TDF domains. Each of the salient TDF domains and the additional theme are presented.

**Table 2** Responses to selected TDF questions by whether the respondent said they involved PPP in numerical aspects of trials – median (percentile 25 – percentile 75), count for scale variables or n (%) out of N for binary variables.

| | Respondents involve PPP (n=116) | Respondents do not involve PPP or are not sure (n=71) |
|---|---|---|
| **TDF domain: knowledge** | | |
| How familiar are you with involving PPP in numerical aspects of trials? | 4 (3.0–5.0), 116 | 2 (1.0–3.0), 71 |
| **TDF domain: skills** | | |
| Grade your own ability to involve PPP in numerical aspects of trials. | 4 (3.0–5.0), 116 | 2 (1.0–4.0), 71 |
| **PPP skills** | | |
| Do you think the PPP need to have particular experience or skills to be involved in numerical aspects of trials? | | |
| Yes | 52 (44.8) | 41 (57.7) |
| No | 64 (55.2) | 30 (42.3) |
| **TDF domain: social/professional role** | | |
| Is involving PPP in numerical aspects of trials an expected role within your job? | 46 (39.7) | 3 (4.2) |
| **TDF domain: beliefs about capability** | | |
| Do you feel confident in your ability to involve PPP in numerical aspects of trials? | 4 (3.0–5.0), 111 | 3 (2.0–4.0), 62 |
| Do you think that involving PPP in numerical aspects of trials is hard to deliver? | 52 (44.8) | 34 (47.9) |
| **TDF domain: beliefs about consequences** | | |
| There is a good balance between the challenges of involving PPP in numerical aspects of trials and the potential benefits. | 68 (58.6) | 22 (31.0) |
| **TDF domain: reinforcement** | | |
| You get recognition from PPP when you involve them in numerical aspects of trials. | 35 (30.2) | 12 (16.9) |
| You get recognition from work peers. | 21 (18.1) | 5 (7.0) |
| You get recognition from your manager. | 23 (19.8) | 6 (8.5) |
| **TDF domain: environmental context and resources** | | |
| Have the resources needed to involve PPP in numerical aspects of trials. | 55 (47.4) | 16 (22.5) |
| Employer provides support to involve PPP in numerical aspects of trials. | 47 (40.5) | 13 (18.3) |
| Employer provides training to involve PPP in numerical aspects of trials. | 31 (26.7) | 7 (9.9) |
| Involving PPP in numerical aspects of trials is compatible with daily practice. | 63 (54.3) | 17 (23.9) |
| PPPs are motivated to get involved in numerical aspects of trials. | 59 (50.9) | 14 (19.7) |
| **TDF domain: social influences** | | |
| Most people who are important think I should involve PPP in numerical aspects of trials. | 5 (4.0–6.0), 94 | 3 (1.0–4.0), 50 |
| Who encourages you to involve PPP in numerical aspects? | | |
| No encouragement. | 7 (6.0) | 14 (19.7) |
| Who is a barrier for you to involve PPP in numerical aspects? | | |
| No barrier. | 47 (40.5) | 13 (18.3) |
| **TDF domain: behavioural regulation** | | |
| Do you have a clear plan on… | | |
| Which numerical aspects of trials you should involve PPP with? | 33 (28.4) | 5 (7.0) |
| How you will involve PPP in numerical aspects of trials? | 36 (31.0) | 6 (8.5) |
| How often will you involve PPP in numerical aspects of trials? | 37 (31.9) | 4 (5.6) |

Data shown as median (25th–75th percentile) and count for scale variables, or n (%) out of N for binary variables.
PPP, patient and public partner; TDF, Theoretical Domains Framework.

## Knowledge: awareness of involving patients and the public in numerical aspects of trials

Respondents who involved public and patient partners in numerical aspects scored higher in the knowledge domain (median 4 vs 2 out of 7 (very familiar with PPI in numerical aspects of trials) for those who did and did not involve public and patient partners in numerical aspects, respectively). Knowledge was highlighted as a barrier to involving public and patient partners in numerical aspects of trials in the open questions where respondents discussed their lack of understanding of how or when to involve public and patient partners and suggesting training as a solution. Respondents highlighted public and patient partners' lack of understanding of numerical aspects or trial processes as a barrier and suggested solutions to overcoming it, such as the use of visual aids or clear language.

**Table 3** Open question responses coded by domain and grouped according to specific beliefs

| Domains | Specific belief | Illustrating quotes |
|---|---|---|
| Knowledge | Lack of clarity about what the task entails | "[What would help me involve PPP in numerical aspects of trials is…] Understanding what involving patients and public in numerical aspects of trials actually means, how it can be done and why it's important." (TM, NI) |
| | | "I think the main issue here is that there are very simple parts of numerical aspects of trials which we would always, and easily, involve PPI in: outcome measure acceptability and monitoring trial progress. I don't know what other aspects might be, so I am imagining a training or knowledge failure on my part." (TM, involves) |
| | Training | "[What would help me involve PPP in numerical aspects is…] training courses/workshops for members of the public on the numerical aspects of trials and also researchers on how to involve them." (HE, NI) |
| | | "This is something that has never really been discussed at work. Nor have I seen any courses about doing this. More guidance is available about writing simply for patients but not specifically about numerical aspects. Some training would be welcome." (TM, involves) |
| | Knowledge of patient and public partners | "[What makes involving PPP in numerical aspects hard is…] Being confident patients have sufficient understanding to be able to contribute. Not sure what could make it easier, just need to be aware this may be a potential issue during initial discussions with focus group members for example." (TM, involves) |
| | | "[What makes involving PPP in numerical aspects hard is…] a general lack of awareness of how trials (and research) work, which makes the initial engagement harder. It's perfectly possible to overcome this though." (Qualitative researcher, involves) |
| Skills | Skills of patient and public partners | "…I would suggest that if you want to involve PPI in this [numerical] aspect then you would have to purposively approach PPI with a certain skill set or advertise for such PPI involvement." (Coinvestigator, NI) |
| | | "It generally quite boring for the general public and they do need a basic level of numeracy and appreciation of research methods." (Statistician, involves) |
| Social/professional role | Professional role | "The grant applicants and senior team often take these decisions during trial design. I never had any impact on these decisions previously." (Qualitative researcher, involves) |
| | | "Numerical roles already 'assigned' to statisticians and clinicians, so as Trial Manager, it can be difficult to enforce PPI involvement." (TM, involves) |
| Beliefs about capability | Capabilities of patient and public partners | "Often numerical aspects are dictated by statistical programs and I think there is a general feeling that patient or public partners cannot grasp them." (TM, NI) |
| | | "[A barrier to involve PPP in numerical aspects of trials is…] The PP themselves not being confident enough to comment." (Coinvestigator, involves) |
| Beliefs about consequences | Impact on trial quality | "Patients can help you work out how to disseminate the results for example, which graphs are easiest, which format of presenting an estimate (risk, number of days etc)." (Statistician, involves) |
| | | "Discussing numerical information to patients helps to distill complex findings into something more simple and acceptable. This helps the researcher to clearer think through what are the results, helps explain information clearly that improves the quality of the trial and will ultimately all benefit the patient." (TM, involves) |
| | Positive impact on patient and public partners | "The PPP can … increase their knowledge and understanding of their condition, new potential treatments, and the process involved in trials which evaluate treatments. This could reduce the information asymmetry between the patient and clinician in the future." (HE, involves) |
| | | "Patients can understand the difficulties in running clinical research and trials. Showing recruitment figures and follow-up data is also beneficial as patients can see the results accumulating and the benefits of taking part in trials." (TM, involves) |
| | Negative impact on patient and public partners | "Makes participation stressful, unless the PP has pre-existing skills in stats or maths." (Coinvestigator, involves) |
| | Extra work/responsibilities for the researcher | "It is more work for the researcher to involve patients/partners." (Statistician, NI) |
| | Uncertainty about impact on trial quality | "I think it could improve transparency of decision making through a trial process. I am less sure that will actually translate into 'better' decisions." (CI, involves) |
| | | "I have answered no impact because I think it is unclear whether quality or the patient partner will get a positive impact. That is a question that needs investigation." (CI, involves) |
| Reinforcement | Lack of incentives | "Having some tangible reward for your investment! It never does anything for your career, while other priorities do." (Statistician, involves) |
| | | "It's not really rewarded or recognised because it is an expected part of study design. All funders we work with consider PPI involvement with trials in general essential." (TM and statistician, involves) |
| Environmental context and resources | Funding | "[What would help me involve PPP in numerical aspects is…] More funding for a PPI expert at our CTU. I am only funded 10% to be the coordinator for all PPI and to do a good job, I'd like to be able to spend more time on my PPI role." (TM, involves) |

Continued

 Goulao B, *et al. BMJ Open* 2021;**11**:e046977. doi:10.1136/bmjopen-2020-046977

**Table 3** Continued

| Domains | Specific belief | Illustrating quotes |
|---|---|---|
| | Time | "I will need training to effectively include the public and the public will need training to effectively contribute to the research. This will take more time and resources that aren't currently provided but if they were this would make it much easier." (HE, NI) |
| | Staff | "[What would help me involve PPP in numerical aspects is…] More money, more trained help. The usual problems." (TM, involves) |
| | | "[What would help me involve PPP in numerical aspects is…] Having a special person dedicated to running PPI work across trials." (TM, involves) |
| | Difficulty accessing willing volunteers | "It's difficult to find public partners for any research, but when it's something that's specifically numerical I think that's an even bigger challenge!" (Qualitative researcher and TM, involves) |
| | | "[What would help me involve PPP in numerical aspects is…] To know how or where to find participants that would be interested in participate." (TM, NI) |
| Social influences | Supportive team environment | "[What would help me involve PPP in numerical aspects is…] More of a group/team effort so that it is not just left to the trial manager to present to PPI members but the trial team as a whole takes on this responsibility." (TM, involves) |
| | | "I think the enthusiasm of the chief investigator to include PPI members is important. If the investigator wants to involve PPI then it will happen but if the investigator is not bothered then it proves much more difficult to include someone." (TM, involves) |
| | Patient and public partners' interest | "I would be happy to share details of the numerical analysis, but have never been asked to do so - they seem satisfied with the summary explanations provided." (TM, NI) |
| | | "PPPs generally shrink back even when asked to comment on numerical aspects of a trial." (Coinvestigator, involves) |
| Behavioural regulation | Guidance | "… If there was clear guidance and tools on how to do it and those tools could be easily integrated into management structures for running large scale studies then I would use them." (CI, involves) |
| | | "[What would help me involve PPP in numerical aspects is…] Guidelines about PPI expectations for review of numerical aspects." (TM, unsure about involvement) |
| | Standard operating procedures | "[What would help me involve PPP in numerical aspects is…] have this implemented into SOPs [Standard Operating Procedures]." (TM, NI) |
| Trial communication culture | Lack of clarity in communication about numerical and statistical aspects of trials | "language and jargon used around sample size calculations etc in design discussions. Stats/methodologists need to be open and able to discuss complex issues openly and simply." (TM, involves) |
| | | "As a patient partner I feel very privileged… But as a retired maths teacher, I am very aware of the anxiety that numbers cause even otherwise very clever people - these anxieties will only be mitigated by more involvement at all levels, including the public." (PPP, involves) |
| | | "If everyone in the research environment explained what the numbers mean clearly to all members of the team - that would make it easier to describe and facilitate involvement from patient or the public." (TM, NI) |

CI, chief investigator; CTU, clinical trial unit; HE, health economist; NI, does not involve PPP in numerical aspects; PPI, patient and public involvement; PPP, patient and public partner; TM, trial manager.

[What would remind me of doing PPI in numerical aspects of trials is…] A better understanding of what they can bring to other numerical aspects (other than oversight committees). (Trial manager, involves)

Visual aids (e.g. graphs) are useful. Someone on hand to answer questions and to help interpret the information provided. (Trial manager, involves)

### Skills: ability or proficiency to involve patients and the public in numerical aspects of trials

Respondents who involved public and patient partners in numerical aspects scored higher in the skills domain (4 vs 3 out 7 (feeling able to involve public and patient partners in numerical aspects) for those who involve and those who do not, respectively). However, the answers to open questions relevant to the skills domain focused on patients and the public's perceived lack of appropriate skills as a cause of concern and barrier to involve them in numerical aspects of trials.

[What makes involving public and patient partners in numerical aspects hard is public and patient partners…] Sufficient numeracy skills to understand simple questions around proportions etc. (Statistician, does not involve)

Half of the respondents believed public and patient partners need to have a set of experiences or skills to be involved in numerical aspects of trials (44% (n=52) of trialists who involve public and patient partners in numerical aspects believed this vs 58% of those who do not (n=41)). When asked to specify what skills public and patient partners would need, some respondents focused on general skills such as lived experience of disease or communication. Other respondents thought public and patient partners had to have technical skills in order to be involved, such as numeracy or mathematical skills, statistical understanding, data interpretation and understanding of graphs, knowledge of trial methodology, trial processes and research in general.

### Social or professional role: perception of own role and responsibilities in relation to involving patients and the public in numerical aspects of trials

Of respondents who involved public and patient partners in numerical aspects, 40% considered it an expected role within their jobs (n=46) vs 4% of those who did not involve public and patient partners (n=3). This was supported by qualitative data that showed that some trialists felt like other colleagues, including the chief investigator or statistician, would have to drive this process.

> [A barrier to involving public and patient partners in numerical aspects of trials is…] It is not spoken about much and does not always relate directly to job roles. (Statistician, not sure about involvement)

### Beliefs about capability: self-belief or confidence in the ability to involve patients and the public in numerical aspects of trials

Beliefs about trialists' own capability did not seem to drive behaviour, with for example a similar percentage of respondents who did (n=52, 45%) and who did not (n=34, 48%) involve public and patient partners finding the task difficult. However, responses to open questions highlighted trialists' beliefs about public and patient partners' capability as a barrier, suggesting public and patient partners feel intimidated by numbers, do not grasp them or do not feel confident enough to comment.

> [Two public and patient partners are involved in sample size considerations and recruitment figures…] Both are already confident with numbers/figures so only needed some support with specifics of presentation/meaning. (Trial manager, involves)

### Beliefs about consequences: beliefs about outcomes of involving patients and the public in numerical aspects of trials

More than half of respondents who involved public and patient partners in numerical aspects believe there is a good balance between challenges and benefits of doing so (n=68), whereas only 30% shared that belief in those who did not involve (n=22). Responses to open questions showed suggestions of positive consequences: focused on the project (such as improving the interpretation of the results, data collection tools and processes, dissemination, trial recruitment, understanding of missing data, interpretation of clinical significance) or the public and patient partners (leads to better understanding of the data and trial processes, more motivated public and patient partners and better team communication). Some respondents suggested negative or no consequences, such as the impact on public and patient partners being unclear or stressful, a disbelief that it can result in improved trial quality, and extra work and responsibilities for the researcher.

> [Involving public and patient partners in numerical aspects of trials can have…] Multiple impacts, most importantly allowing trial teams to best understand how the key messages of a study can be translated directly to those

patients living with condition under study. (Statistician, involves)

### Reinforcement: whether there is external stimulus to involve patients and the public in numerical aspects of trials

A third of respondents believed there was recognition from public and patient partners when involving them in numerical aspects of trials if they did so already (n=35) vs 17% of respondents who did not (n=12). Data from open questions supported these observations, with trialists mentioning a lack of tangible rewards or encouragement as barriers to involving public and patient partners in numerical aspects.

### Environmental context and resources: factors related to the work setting that influence involving patients and the public in numerical aspects of trials

Less than half of trialists felt like they had the resources to support public and patient involvement in numerical aspects if they were doing it (n=55, 47%) and only 23% (n=16) if they were not. This was illustrated in several replies to the open questions, with respondents suggesting more resources (financial, human) could facilitate involvement, as well as support identifying interested public and patient partners.

### Social influences: perception of how others see one's role and how it impacts on the ability to involve patients and the public in numerical aspects of research

Of those who involve public and patient partners in numerical aspects, 94% believe there is some type of encouragement from others to do so (n=7) vs 80% of those who do not involve public and patient partners (n=14). Of those who involve public and patient partners, 40% believe that no one acted as a barrier to do the target behaviour (n=47) vs 18% of those who do not involve public and patient partners (n=13). Qualitative data showed support from colleagues as a team effort and encouragement from management or from public and patient partners were important factors in involving them in numerical aspects of trials.

> Operationally, I don't think PPI members really add a great deal. I also don't think they tend to be interested in those [numerical] aspects. This may be because of the public perception of data and statistics as being hard/obtuse/esoteric. (Trial manager and statistician, involves)

### Behavioural regulation: perception of guidance or tools to manage or change PPI in numerical aspects of trials

About a third of those who involve public and patient partners in numerical aspects state they have a clear plan about which aspects to select, how and when, compared with around 7% of those who do not involve public and patient partners. Replies to open questions showed trialists suggest the need for clear guidance on how to conduct PPI in numerical aspects could help facilitate the behaviour, as well as having it as a requirement on ethics committees or standard operating procedures.

> …REC [Research ethics committee] could remind when providing the final report to consider this. The Funder

often requires a report so they too could advise to do this. Being reminded by bodies that you have a regulatory or contractual obligation would help. May be journals could advise that this is done in their Instructions to Authors. The Research Design Service (RDS) also has PPI representatives on their teams and they could act in a capacity to remind people to do this. Internally in my unit, we could do with a guidance document about this. (Trial manager, involves)

### Trial communication culture

Several comments from trialists talked to a broader issue with trial communication culture which highlighted the challenges with communicating clearly about numerical aspects with the entire trial team and ensuring everyone, including patient partners, fully understand them.

Numbers are presented to the PPI partners all the time in every trial... So they are involved, but it is by no means actually clear if they or indeed anybody else on the trial team actually understands the figures and the way they are presented. Are they actually any different from anybody else on the trial team? Would improve transparency of decision making which I think would be an improvement even if the actual decisions within the study do not qualitatively change. (Chief investigator, involves)

### DISCUSSION

We have conducted a large UK-based survey to identify current practice, barriers and facilitators of involving public and patient partners in numerical aspects of trials . Because numerical aspects are central to the design, conduct, analysis and dissemination of trials, our study explicitly asks about involvement in such aspects rather than more broadly. Our sample represented a wide range of ages, professions and geographical regions in the UK. We used a theory-informed approach to design and analyse our findings to identify barriers and facilitators to involvement of public and patient partners in numerical aspects of trials. We found, in general, trialists believe involving public and patient partners in numerical aspects of trials is a good thing to do, but reported several barriers to doing it.

Around 60% of respondents to the survey reported to involve public and patient partners in the numerical aspects of trials; a similar percentage of women (62%) reported involving public and patient partners compared with men (66%). There was a lower percentage of early career trialists (up to 5 years of experience) doing this (49%) compared with more experienced trialists (72% and 63% for trialists with 5–10 years of experience and over 10 years, respectively). This contrasts with the report that female and junior researchers are often tasked with doing PPI in health research,[3] although it could be an artefact of respondents who are particularly interested in PPI. There is a higher proportion of trialists involving public and patient partners in surgical trials (92%[4]) or even in all trials, 10 or more years ago (75%[5]), than the

proportion we found involving public and patient partners in numerical aspects. Most of our respondents involved public and patient partners in numerical aspects at the trial design (68%) or dissemination (63%) stages, with a significant percentage of respondents involving public and patient partners in undertaking the trial (44%) and in its analysis (35%). Crocker et al's survey of PPI in surgical trials[4] and Raza et al's audit of trials in the Integrated Research Application System[13] showed very similar patterns, although both observed a lower prevalence of involvement in the undertaking of a trial. Since involving patients and public partners in oversight committees is common practice[2] but was not a stand-alone option in our survey, it is possible respondents included it as part of trial conduct.

There is scope to improve trialists' knowledge, environmental context and resources, and behavioural regulation regarding involving public and patient partners in numerical aspects of trials. Common reflections from the open questions revealed the need for training, more funding and resources, such as finding public and patient partners that are interested in being involved in numerical aspects of trials; developing guidelines about how and when public and patient partners can be involved in numerical aspects would be a natural next step. Trialists have also reported low recognition from public and patient partners, work peers and their managers. These findings are in line with other studies focused on general PPI in research[3 14] and with wider PPI training needs requested by researchers working in clinical trials in the UK.[15] However, this need was bigger in trialists who did not involve public and patient partners in numerical aspects of trials.

There was a clear difference in the perceived balance of benefits versus challenges of PPI in numerical aspects of trials between those who reported they did it (over 50% reported it as a positive balance) and those who did not (around 30% said the same). This is expected as professionals enacting a behaviour will tend to perceive more benefits than challenges compared with those who are not doing it. Respondents of the survey reported more accessible and implementable findings, the opportunity of questioning researchers, and more transparent partnerships between researchers and public and patient partners as benefits of involving public and patient partners in numerical aspects of trials. However, some respondents saw this as an extra time-consuming task with no evidence of improving the decision making in trials and/or having an impact of patient partners. It is possible these barriers appear more salient to those not involving patient and public partners in numerical aspects and the reasons for this can be varied, including their own workplace culture. This is reflected in numerous other studies about researchers' motivations to involve patients and the public as partners in research and the ongoing debate about measurable impact.[3 16]

It is unclear who should be responsible for the involvement in numerical aspects of trials, given chief investigators, statisticians and health economists traditionally have the role of analysing or sharing trial data and results, but trial managers are more often in charge of delivering PPI. This perception was echoed in the qualitative data where junior researchers,

trial managers and qualitative researchers suggested it was not their responsibility or it was difficult for them to involve public and patient partners in numerical aspects of trials. This was clearly an important factor in the respondents' behaviour, since only three respondents (4%) considered involving public and patient partners in numerical aspects of trials as part of their role from trialists who were not currently performing the behaviour (vs 40% of those who did it).

There is uncertainty about public and patient partners' motivation to get involved in numerical aspects of research. Around half of the respondents who involved public and patient partners in numerical aspects and 20% of trialists who do not perceived public and patient partners as being motivated to get involved. The thematic analysis identified trialists perceived patient partners' lack of interest or finding numbers 'boring' as a barrier to involving them in numerical aspects of trials. Our survey included only four public and patient partner respondents and therefore cannot draw conclusions from their perspective, but there is scope for more research understanding public and patient partners' expectations when it comes to involvement in numerical aspects of trials. However, it seems unlikely trialists could confidently speak on behalf of their public and patient partners in terms of motivation to get involved in specific trial aspects. This is because a key challenge raised by several open question comments is the lack of definition of expectations about public and patient partners from the start. Defining expectations is not common practice, even though it is considered useful by all stakeholders.[4 15]

Tokenism and the idea that public and patient partners can only contribute to a limited number of tasks related to increasing readability may be a significant barrier to involving public and patient partners in numerical aspects of trials. A lot of respondents perceived numeracy and statistical knowledge as essential for public and patient partners to get involved in numerical aspects of randomised controlled trials and, in some cases, advanced knowledge about sample size calculations and analysis. This contrasts with the fact that public and patient partners often get involved in qualitative research and reinforces perceptions of disciplinary hierarchies,[3] but could also be related to the fact that PPI and qualitative research have many common features and therefore seem a natural crossover.[17] In fact, Boylan et al[3] reported an assumption from UK researchers that the involvement of patients in qualitative interviewing and analysis is uncontroversial; some of our respondents have a clearly different view when it comes to public and patient partners' involvement in quantitative analysis. From the public and patient partners' perspective, a respondent pointed out that they often felt underestimated and another that tokenism is the main barrier to involving public and patient partners in numerical aspects. Wider literature on the topic shows there are appropriate methods to involve non-experts in discussions about statistical models and numbers,[6] although this is a controversial topic.[7] Either way, communication, in particular about numerical aspects of trials, needs to improve to enable meaningful PPI in trials, and this has been known for almost a decade[5] and it is a key priority for future PPI research according to

multiple stakeholders.[18] A way forward could, for example, follow models to achieve better communication related to patient data, such as the Understanding Patient Data project (https://understandingpatientdata.org.uk/).

### Limitations

Like with most complex behaviours, involving public and patient partners in numerical aspects of trials is in a spectrum. Respondents to our survey interpreted it in a variety of ways, ranging from presenting numbers in reports to public and patient partners at trial steering meetings to involving them in doing the statistical analysis. Despite the iterative development of a definition endorsed by multiple trial stakeholders and its presentation at the beginning of the survey, it is likely that the barriers and facilitators presented here regard a range of behaviours in that spectrum. To develop the survey, our PPI was limited to piloting. Since the survey aimed to understand practices, barriers and facilitators for professionals, we considered this to be sufficient within the context of a master's student's project, but patient and public partners' perspective could have enriched the study questions and survey design and potentially increased the number of patient and public partners responding. We did not include a stand-alone option of PPI in the numerical aspects of data and trial steering committees; however, we do not consider this to be essential information to our purpose (a broad overview of PPI in numerical aspects of trials), and nevertheless respondents added this information in open questions. Qualitative analysis was based on open questions in a survey which are limited in their scope when compared with conducting interviews and may lead to misinterpretation of answers.[19] We tried to minimise this by double assessing a subsample of qualitative data. Our sample was diverse and came from multiple locations and professions across the country. Respondents who answered demographic questions but did not proceed to answering questions specifically about PPI in numerical aspects of trials were mostly similar to those who did; however, we are likely to have respondents who have stronger views of PPI and PPI in numerical aspects of trial since they opted to answer our survey.

### CONCLUSION

We found a generally positive attitude towards involving public and patient partners in numerical aspects of trials, with 60% of the respondents reporting attempting to do so in a variety of ways. However, there is a need to understand public and patient partners' expectations when getting involved in a trial and whether they would have an interest in being involved in numerical aspects. To achieve this, public and patient partners first need to be aware of the different stages of trials and the opportunities to get involved. Numerical aspects are a key element in trials and therefore should at least be part of an initial discussion. Our survey has also highlighted that communication of numerical and methodological aspects of trials can be challenging, not just for public and patient partners but for the whole trial team. Improving it should be

a matter of providing training on either communication skills, technical aspects or both, and it should be available for the whole team, including but not limited to public and patient partners. A training set codesigned by relevant stakeholders and available to all clinical trials units in the UK, including successful case studies of involving public and patient partners in numerical aspects of trials, is an essential step forward, as well as guidance on what are the most relevant numerical aspects of trials to involve public and patient partners in and how to do that in a meaningful way.

**Acknowledgements** We would like to thank the staff and student members at the University of Aberdeen who helped pilot-test the survey, and in particular Dr Eilidh Duncan for her expertise and support in the development of the questions underpinned by the Theoretical Domains Framework. We would like to acknowledge all trialists who took the time to reply to this survey and everyone who helped disseminate it.

**Contributors** CP wrote the first draft of the protocol. BG and KG contributed to the design of the project and development of the protocol. BG and CP conducted the quantitative analysis. BG conducted the qualitative analysis. KG checked a sample of BG's analysis as a quality check. BG wrote the first of the manuscript. All authors approved the final version of the manuscript.

**Funding** BG was supported to develop this research by the Wellcome Trust Institutional Strategic Support Fund at the University of Aberdeen.

**Competing interests** None declared.

**Patient consent for publication** Not required.

**Ethics approval** We conducted an online survey approved by the College of Ethics Research Committee (CERB/2019/4/1759) of the University of Aberdeen.

**Provenance and peer review** Not commissioned; externally peer reviewed.

**Data availability statement** Data are available upon reasonable request.

**ORCID iDs**
Beatriz Goulao http://orcid.org/0000-0003-1490-7183
Katie Gillies http://orcid.org/0000-0001-7890-2854

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
