## [Reviewer comments · BMJ Open]

ARTICLE DETAILS

TITLE (PROVISIONAL)	Patient and public involvement in numerical aspects of trials: a mixed methods theory-informed survey of trialists' current practices, barriers and facilitators
AUTHORS	Goulao, Beatriz; Poisson, Camille; Gillies, Katie

VERSION 1 – REVIEW

REVIEWER	Bec Hanley I'm a freelancer. I work for a few hours each week for the Medical Research Council Clinical Trials Unit at UCL in the UK
REVIEW RETURNED	17-Dec-2020

GENERAL COMMENTS	This is a really interesting topic - thank you for working on it and for submitting this article for publication. All of my feedback should be read in the context of this sentence. I have used the page numbers at the bottom of the page – not the ones at the top. General comments Was there PPI in this project? I couldn't see any evidence of it (e.g. not listed in the list of people involved in the development of the survey on page 4) and didn't see a PPI contributor listed as a co-author. If there wasn't any PPI, it would be helpful to say this and to include it as a limitation. I wasn't clear about why you asked particular questions – e.g. about gender, age, geography – what conclusions did you hope to draw from these? Does it matter that 70% of respondents were women? Why? What's the significance of age? The English felt a bit clunky in some places and would benefit from a more detailed proof read – e.g. page 9 line 40 “around 60% of respondents reported to involve...”; page 10 line 27 “This was a clearly important factor for responders to perform the behaviour...” I'd suggest adding a clearer section on limitations Detailed comments Page 3 line 7/8 – might be helpful to reference this statement Page 4 line 21/22 – I couldn't see the questionnaire in the papers that I could access – it would be helpful to see it Page 4 line 42/43 – Do the authors think that the fact that the survey was only open for a relatively short time during the summer had any impact on the number of responses? If so it may be worth mentioning this in the discussion section. Page 5 – results paragraph. I got very confused by the numbers here. I think what you're saying is that if people only replied to the 1st 3 sets of questions listed in the bullet points on page 3, you are describing them as non-respondents in this paragraph and the rest of the paper? If so, where are the results from people who
--

	answered the general questions about PPI in numerical aspects of trials but didn't answer the TDF questions? If not, can you be clearer about your definitions? Page 6, lines 13-15. What kind of comments were received? Page 6, lines 56 – 58. I was disturbed by this sentence. It feels like the authors have taken the responses from the survey and repeated them without applying any qualifying statements. So I'd rather this sentence said "However, the answers to open questions relevant for skills focused on patients' and publics' perceived lack of appropriate skills....." Page 7 lines 3-4. Again I was disturbed by this quote being included without any commentary. How is lack of current affairs knowledge even relevant here? Page 7 line 49. I felt this was a crucial statement and I'd like to see it discussed more in the discussion section. Page 9 line 51. Was the fact that you didn't have a stand-alone question about membership of trial oversight committees a limitation? Page 10 line 39. Only 4 PPI contributors responded, maybe because the questions asked things about working in a Unit for example. I'd make this bit a part of a limitations section.
--	--

REVIEWER	Simon Denegri No affiliation Executive Director of the Academy of Medical Sciences (salaried) I am also Chair of the UKCRC CTU Network Executive Group (unpaid)
REVIEW RETURNED	20-Dec-2020

GENERAL COMMENTS	The degree to which public involvement takes place in numerical aspects of trials is an important question. But very few studies have been conducted in this area to my knowledge. The article will be valuable to BMJ readers in beginning to explore attitudes and approaches and identifying current tensions that may present as barriers to involvement. It certainly merits follow-up with further studies that look in more detail at attitudes among individual constituencies (patients and carers, Chief Investigators etc) where the sample sizes are greater. It also contains many learnings for consideration by trialists, clinical trials units and funders. and the authors identify a number of potentially useful actions that could flow. In the discussion the authors rightly identify themes around public understanding and communication of data that are common to other areas of science (AI, health data generally). Further references to works in this area might be helpful or to the work of relevant organisations such as Understanding Patient Data. I would be interested to know whether any public members were involved in this study, it's design, conduct and analysis? The irony of not doing so given the nature of the study may not be lost on some readers.
--

VERSION 1 – AUTHOR RESPONSE

Reviewer #1:

3	Was there PPI in this project? I couldn't see any evidence of it (e.g. not listed in the list of people involved in the development of the survey on page 4) and didn't see a PPI contributor listed as a co-author. If there wasn't any PPI, it would be helpful to say this and to include it as a limitation. Response We have made this clearer by adding a section on patient and public involvement in the study, in the methods section, page 5: "Patient and public involvement in the study Our study is part of a larger project, the PoINT project, where patient and public involvement has been sought at all trial stages including in design and conduct. The current survey was piloted by three public members." We have also added it to the discussion section of the paper under the limitation sub-heading (please see response to comment 16).
4	I wasn't clear about why you asked particular questions – e.g. about gender, age, geography – what conclusions did you hope to draw from these? Does it matter that 70% of respondents were women? Why? What's the significance of age? Response We have included these characteristics as a way of describing the sample that replied to our survey. In addition, we were interested in assessing how our sample compared to others previously reported to be engaged in PPI activities. To reflect this, we have added a sentence to the discussion: "a similar percentage of women (62%) reported involving public and patient partners compared with men (66%); there was a lower percentage of early career trialists (up to 5 years of experience) doing this (49%) compared with more experienced trialists (72 and 63% for trialists with 5-10 years of experience and over 10 years, respectively). This contrasts with the report that female and junior researchers are often tasked with doing PPI in health research (Boylan), although it could be an artifact of responders that are particularly interested in PPI."
5	The English felt a bit clunky in some places and would benefit from a more detailed proof read – e.g. page 9 line 40 "around 60% of respondents reported to involve..."; page 10 line 27 "This was a clearly important factor for responders to perform the behaviour..." Response

	Thank you for pointing this out. We have made changes to the wording to make the manuscript clearer.
6	I'd suggest adding a clearer section on limitations Response We have added information to the discussion section and, in particular, the limitations paragraph in page 11, which now has a clear sub-heading. In our responses below, we specify the information added to the limitation paragraph.
7	Detailed comments Page 3 line 7/8 – might be helpful to reference this statement Response We have not added a reference as we were unsure what statement this refers to – the statements we see on lines 7 and 8 are referenced.
8	Page 4 line 21/22 – I couldn't see the questionnaire in the papers that I could access – it would be helpful to see it Response The questionnaire is now available in Appendix 1.
9	Page 4 line 42/43 – Do the authors think that the fact that the survey was only open for a relatively short time during the summer had any impact on the number of responses? If so it may be worth mentioning this in the discussion section. Response The literature on online survey methods indicates that most replies are received within the first few days of receiving an invitation (Tai et al, BMC Medical Res Meth 2018. 18 (59); Fan Computers in Human Behaviour 2010. 26 (2): 132-139). We do not believe the timing impacted on response rates and given the survey was closed before the main British School holidays, we believe most people would have had the opportunity to contribute.

10	Page 5 – results paragraph. I got very confused by the numbers here. I think what you're saying is that if people only replied to the 1st 3 sets of questions listed in the bullet points on page 3, you are describing them as non-respondents in this paragraph and the rest of the paper? If so, where are the results from people who answered the general questions about PPI in numerical aspects of trials but didn't answer the TDF questions? If not, can you be clearer about your definitions? Response This is correct. We have edited the text to make this clearer: "A total of 317 people accessed the survey. From those 317, we excluded from the final analysis 130 people for the following reasons: 31 did not provide any information; 99 did not respond to any TDF focussed questions. Therefore, the 187 participants that replied to at least one TDF question were included in the final analysis (Figure 1)". The replies from people who answered general questions but did not answer the TDF questions are in Supplementary table 1. This is indicated in the results section of the manuscript "Supplementary table 1 provides demographic information based on respondent status."
11	Page 6, lines 13-15. What kind of comments were received? Response We have added an example to illustrate our finding in page 6, lines 13-15: "(for example, "I've also yet to have any PPI members say they would like to be more involved in this side of things; except for post-analysis when considering the qualitative and quantitative aspects together")."
12	Page 6, lines 56 – 58. I was disturbed by this sentence. It feels like the authors have taken the responses from the survey and repeated them without applying any qualifying statements. So I'd rather this sentence said "However, the answers to open questions relevant for skills focused on patients' and publics' perceived lack of appropriate skills....." Response We agree with this comment and have made the appropriate change to the sentence.
13	Page 7 lines 3-4. Again I was disturbed by this quote being included without any commentary. How is lack of current affairs knowledge even relevant here?

	Response We understand the sentence used to illustrate this domain might be controversial, so we have selected a different sentence instead: “Sufficient numeracy skills to understand simple questions around proportions etc”
14	Page 7 line 49. I felt this was a crucial statement and I'd like to see it discussed more in the discussion section. Response We have added the following to the discussion section (p10) to highlight this finding: “There was a clear difference in the perceived balance of benefits versus challenges of PPI in numerical aspects of trials between those that reported they did it (over 50% reported it as a positive balance) and those that reported they did not (around 30% said the same). This is expected as professionals enacting a behaviour will tend to perceive more benefits than challenges compared with those that are not doing it. Responders to the survey reported more accessible and implementable findings, the opportunity of questioning researchers and more transparent partnerships between researchers and public and patient partners as benefits of involving public and patient partners in numerical aspects of trials. However, some responders saw this as an extra time-consuming task with no evidence of improving the decision making in trials and/or having an impact of patient partners. It is possible these barriers appear more salient to those not involving patient and public partners in numerical aspects and the reasons for that can be varied including their own workplace culture. This is reflected in numerous other studies about researchers’ motivations to involve patients and the public as partners in research and the ongoing debate about measurable impact [3], [16]. “
15	Page 9 line 51. Was the fact that you didn't have a stand-alone question about membership of trial oversight committees a limitation? Response Even though PPI in numerical aspects of trial steering committees might be potentially under-reported here due to the lack of a stand-alone option to report it, we considered this to be a part of trial conduct and we did not aim to investigate it specifically. We also observed responders used open questions to add information about this type of involvement despite a discrete option. We added the following sentences to p11, discussion section to highlight these reflections: “We did not include a stand-alone option of PPI in numerical aspects of data and trial steering committees; however, we do not consider this to be essential information to our purpose (a broad overview of PPI in numerical aspects of trials) and, nevertheless, respondents added this information in open questions”
16	Page 10 line 39. Only 4 PPI contributors responded, maybe because the questions asked

	things about working in a Unit for example. I'd make this bit a part of a limitations section. Response We have now added this as a limitation along with the lack of in-depth patient and public involvement in the design of the project / survey: "To develop the survey, our patient and public involvement was limited to piloting. Since the survey aimed to understand practices, barriers and facilitators for professionals, we considered this to be sufficient within the context of a Master's student project, but patient and public partners' perspective could have enriched the study questions and survey design and potentially increased the number of patient and public partners responding. "
Reviewer #2	
17	The degree to which public involvement takes place in numerical aspects of trials is an important question. But very few studies have been conducted in this area to my knowledge. The article will be valuable to BMJ readers in beginning to explore attitudes and approaches and identifying current tensions that may present as barriers to involvement. It certainly merits follow-up with further studies that look in more detail at attitudes among individual constituencies (patients and carers, Chief Investigators etc) where the sample sizes are greater. It also contains many learnings for consideration by trialists, clinical trials units and funders. and the authors identify a number of potentially useful actions that could flow. In the discussion the authors rightly identify themes around public understanding and communication of data that are common to other areas of science (AI, health data generally). Further references to works in this area might be helpful or to the work of relevant organisations such as Understanding Patient Data. Response Thank you for your suggestion. We were unaware of the Understanding Patient Data project and agree this is a great example of how to improve communication about data with patients. We have added this to our discussion, page 11: "Either way, communication, in particular about numerical aspects of trials, needs to improve to enable meaningful PPI in trials and this has been known for almost a decade [5] and it is a key priority for future PPI research according to multiple stakeholders [18]. A way forward could, for example, follow models to achieve better communication related to patient data, such as the Understanding Patient Data project (https://understandingpatientdata.org.uk/)."
18	I would be interested to know whether any public members were involved in this study, it's design, conduct and analysis? The irony of not doing so given the nature of the study may not be lost on some readers. Response Public partners were involved in piloting the survey. The survey is part of a larger project (PoINT – Patient and public Involvement in Numerical aspects of Trials) and we have patient and public input in that project, but we have not sought any further involvement in the design

	of our survey. We clarified this in the methods section (please see response to #3) and in the discussion (please see response to #16).
--	---

VERSION 2 – REVIEW

REVIEWER	Bec Hanley MRC CTU at UCL UK
REVIEW RETURNED	15-Feb-2021

GENERAL COMMENTS	Thank you for addressing my comments so thoroughly.
---

REVIEWER	Simon Denegri Academy of Medical Sciences, UK
REVIEW RETURNED	20-Feb-2021

GENERAL COMMENTS	Many thanks for considering our comments and amendment the paper accordingly. I think it is a helpful piece of work.
--